# Cobalt Ferrite Magnetic Nanoparticles for Tracing Mesenchymal Stem Cells in Tissue: A Preliminary Study

**DOI:** 10.3390/ijms23158738

**Published:** 2022-08-05

**Authors:** Željka Večerić-Haler, Nika Kojc, Karmen Wechtersbach, Martina Perše, Andreja Erman

**Affiliations:** 1Department of Nephrology, University Medical Center Ljubljana, 1000 Ljubljana, Slovenia; 2Faculty of Medicine, University of Ljubljana, 1000 Ljubljana, Slovenia; 3Institute of Pathology, Faculty of Medicine, University of Ljubljana, 1000 Ljubljana, Slovenia; 4Medical Experimental Centre, Faculty of Medicine, University of Ljubljana, 1000 Ljubljana, Slovenia; 5Institute of Cell Biology, Faculty of Medicine, University of Ljubljana, 1000 Ljubljana, Slovenia

**Keywords:** mesenchymal stem cells, metal nanoparticles, markers, tissue injury

## Abstract

Therapy with mesenchymal stem cells (MSCs) is promising in many diseases. Evaluation of their efficacy depends on adequate follow-up of MSCs after transplantation. Several studies have shown that MSCs can be labeled and subsequently visualized with magnetic nanoparticles (NPs). We investigated the homing of MSCs labeled with magnetic cobalt ferrite NPs in experimentally induced acute kidney injury in mice. To explore the homing of MSCs after systemic infusion into mice, we developed a pre-infusion strategy for optimal tracing and identification of MSCs with polyacrylic acid-coated cobalt ferrite (CoFe_2_O_4_) NPs by light and transmission electron microscopy (TEM) in various organs of mice with cisplatin-induced acute kidney injury and control mice. By correlative microscopy, we detected MSCs labeled with NPs in the lungs, spleen, kidney, and intestine of cisplatin-treated mice and in the lungs and spleen of control mice. Our results confirm that labeling MSCs with metal NPs did not affect the ultrastructure of MSCs and their ability to settle in various organs. This study demonstrates the usefulness of cobalt ferrite NPs in ex vivo visualization of MSCs and offers correlative microscopy as a useful method in routine histopathology laboratories for tracing MSCs in paraffin-embedded tissue.

## 1. Introduction

Mesenchymal stem cells (MSCs) are a heterogeneous population of adult stem cells with the ability to self-renew and to differentiate along different cell lineages. They are particularly abundant in bone marrow, the most common origin of these cells, but they are also present in adipose tissue, cord blood, umbilical cord, and several other tissues of adult human organs [1]. Due to their potential to regenerate and repair diseased and damaged tissues, their use has been extremely expanded in the last decade, offering promising solutions for the treatment of diseases and injuries that cannot be effectively addressed with conventional therapies. To date, MSCs have been studied in the regeneration of various human tissues and organs, such as cardiovascular tissue, spinal cord, bone, cartilage, lungs, as well as in therapy of various autoimmune diseases, diabetes, and neurological and kidney diseases [2].

The therapeutic capabilities of MSCs are thought to be based on their ability to secrete a number of immunoregulatory molecules, cytokines, and growth factors with antiapoptotic and proangiogenic properties, as well as their ability to reduce scarring and inflammation. Although these effects may not be due to the transdifferentiation of transplanted MSCs but mainly to endocrine and paracrine mechanisms, the bioavailability of MSCs at the site of injury appears to be one of the most important factors in achieving optimal therapeutic potential [3,4]. However, although the therapeutic potential of MSCs has been investigated in numerous clinical trials (for details, see http://clinicaltrials.gov (accessed on 15 March 2022)), few of them have been successfully translated into regular clinical practice [4]. This is partly due to the risks and side effects posed by MSCs after transplantation, but is also related to the success of pre-delivery labeling procedures that subsequently affect MSCs detection, homing, and therapeutic potential [5].

To study and better understand the homing process, survival, distribution, and engraftment of exogenously infused MSCs, accurate tissue tracking methods are essential. Over the past decade, non-invasive imaging technologies based on magnetic nanoparticles (NPs) have been developed to track transplanted MSCs in vivo [6]. Several studies have shown that stem cells can be labeled with superparamagnetic particles prior to their transplantation and detected in vivo by magnetic resonance imaging (MRI) [7,8]. Iron oxide NPs are the most widely accepted particles due to their excellent superparamagnetic properties [7]. However, their routine use has not yet found its way into the clinical pipeline due to growing concerns about their toxicity [9]. One important strategy to combat these issues is to dope iron oxide NPs with other metallic elements that have magnetic or other imaging properties, such as cobalt (Co), to reduce the amount of iron released into the bloodstream or other tissues [9]. The addition of a second metal must not cause additional toxicity but must preserve the ability to detect the grafted MSCs, their homing, and their therapeutic potential. These actions lead to the formation of NPs with interesting properties that offer the possibility of combining different imaging modalities to develop more effective and complete diagnostic tests.

In the present study, MSCs were labeled with superparamagnetic cobalt ferrite NPs (CoFe_2_O_4_), which are non-toxic in the short-term and stable in physiological conditions [10], and, to our knowledge, administered for the first time to mice with experimentally induced severe kidney injury. The aim of this study was to investigate the potential of CoFe_2_O_4_ NPs as markers for the visualization of MSCs in tissues (by light microscopy and transmission electron microscopy (TEM)) in order to provide additional information about the morphology and location of MSCs in different organs beyond that obtainable by MRI-based tracking, which is currently the primary indication for nanoparticle-labeling.

## 2. Results

On the fourth day after induction of cisplatin injury the mice showed signs of systemic toxicity, including neurotoxicity, nephrotoxicity, gastrointestinal toxicity, and severe myelosuppression. As detailed in our previous studies [11,12].

### 2.1. Histopathological Characteristics of Cisplatin-Induced Tissue Injury

In the spleens of cisplatin-treated mice (Figure 1A), widening of the red pulp was noted by light microscopy, and the boundaries between white and red pulp were blurred. In the red pulp, there were numerous erythrocytes, as well as numerous polymorphonuclear cells such as neutrophilic granulocytes and macrophages with golden-brown pigment hemosiderin (confirmed by Perl staining).

In the renal cortex of cisplatin-treated mice (Figure 1B), numerous apoptotic and necrotic tubular epithelial cells were found, along with very few mononuclear cells in the interstitium and tubules of the renal cortex and medulla. The glomeruli appeared unremarkable.

Analysis of the intestines of cisplatin-treated mice (Figure 1C) revealed numerous apoptotic bodies in the epithelial cells (enterocytes) and necrotic epithelial cells, particularly in crypts. The lamina propria showed few neutrophilic granulocytes. Other layers of the intestinal wall, including the peritoneum, were normal.

Histologic analysis of the lungs from cisplatin-treated mice revealed very few neutrophils and mononuclear inflammatory cells in the interstitium, without other signs of tissue damage.

In the liver of cisplatin-treated mice, there were very few apoptotic cells in either group and no evidence of hepatocyte damage in the cisplatin-treated mice; in the sinusoids, only scattered mononuclear cells, including Kupffer cells, were found in both groups.

### 2.2. MSCs with Nanoparticles Were Visible in Histological, Semi-Thin, and Ultrathin Sections

By light microscopy, scattered cells with brown particles in the cytoplasm were detected in tissues and cells (Table 1). In the lungs, cells with brown particles were located in the interstitium around the capillaries (Figure 2A). In the spleen, cells with brown particles were found predominantly in the red pulp and at the border between red and white pulp, surrounded by erythrocytes, lymphocytes, neutrophilic granulocytes, and scattered plasma cells. In the liver, cells with suspicious brown particles were visible in the cells in the sinusoids. Cells with brown particles were also detected in semi-thin sections in the lungs and spleen (Figure 2B). Detailed examination with TEM revealed that the cells with brown particles detected in the histological and semi-thin sections in the lungs and spleen were MSCs with internalized NPs in endosomes within their cytoplasm (Figure 2C–F). In the liver, cells with suspicious brown particles located in the sinusoids were characterized as Kupffer cells by TEM in both groups.

Using TEM, we identified MSCs in various tissues by the presence of electron-dense NPs in the cytoplasm (Table 1). These cells exhibited typical ultrastructural features of MSCs, such as a centrically located large nucleus with predominant euchromatin and prominent nucleolus, and unabundant cytoplasm with a moderate quantity of organelles (Figure 2C–E and Figure 3A,B).

### 2.3. The Presence of MSCs in Kidney and Intestine Might Be Related to Cisplatin-Induced Injury

MSCs with internalized NPs were found by TEM in kidney and intestine with cisplatin-induced tissue damage in cisplatin-treated mice but not in kidney and intestine without organ damage in control mice. In the lungs and spleen, similar amounts of MSCs were found by light microscopy and TEM regardless of cisplatin treatment. No MSCs were found in the liver regardless of cisplatin treatment.

Electron-dense NPs were seen in endosomes in proximal tubule epithelial cells in the kidney and in endosomes of Kupffer cells in the liver, independent of cisplatin treatment (Figure 3C–F).

### 2.4. MSCs in Tissues Were Additionally Confirmed by Immunolabeling

By double immunolabeling of the surface markers CD44 and CD105, we also confirmed the presence of the injected MSCs in the mouse tissues examined. We could only identify cells with a positive immunoreaction for both surface markers as MSCs (Figure 4).

## 3. Discussion

We have previously shown that extrarenal MSCs contribute to renal repair after acute cisplatin-induced injury and improve animal survival, renal function, and renal tubule epithelial cell recovery [12]. Several cells have been shown to settle in the renal interstitium near the injured tubular epithelia, and we hypothesized that they exert their beneficial effects by reducing tubular cell apoptosis in a paracrine manner. However, previous studies, including ours [13,14], have mainly used fluorescence imaging to follow the cells at the histological level, so we could not draw conclusions about possible transdifferentiation-dependent or other mechanisms of action or the fate of the transplanted MSCs on this basis. Fluorescence microscopic methods indeed have significant limitations, because we cannot determine the exact localization of the labeled cells in the target tissue relative to other cells, nor can we draw conclusions about changes in the morphology of the transplanted cells and their fate on this basis. Moreover, with conventional fluorescence imaging (as with DiI labeling in our previous research), cell imaging and tracking are based on fluorescent dye molecules, which are restricted to a broad emission spectrum, limited emission per molecule, and unsatisfactory photostability [13].

In the present pilot study, we investigated the engraftment of MSCs in an animal model of cisplatin-induced severe acute kidney injury compared with control mice without cisplatin treatment. MSCs were previously labeled in vitro with magnetic cobalt ferrite NPs in a controlled manner. We aimed to use internalized NPs as markers for MSCs, which would help us to detect and recognize MSCs in animal tissues.

In recent years, NPs have become a subject of intense research, especially in the biomedical field, in which magnetic NPs are among promising tools with uses as MR-contrast enhancers, for drug delivery, and for the hyperthermic treatment of tumors [15,16]. Cobalt ferrite NPs appear to overcome many of the aforementioned drawbacks of fluorescent cell death. Cobalt ferrite NPs have been shown to be internalized into the cytoplasm mainly by macropinocytosis and clathrin-mediated endocytosis and to remain in endosomes without damaging the cells [10]. Because of their metallic nature, we could already identify them by light microscopy in classical histological sections as numerous brown particles observed in scattered cells. However, light microscopic analysis allowed the detection of NPs and/or cells containing NPs only in the spleen and lungs of both control and cisplatin-treated animals. We assume that this is probably due to the expected highest density of MSCs in both organs, which has also been observed by other authors [17]. Indeed, during the initial period after the systemic infusion of MSCs, a considerable portion of the administered MSCs are passively trapped in the organs because of their size (e.g., in the lungs) or the filtering function of the organ (e.g., in the liver and red pulp in the spleen). Therefore, only a small fraction of administered MSCs enter diseased tissues, where they are presumably attracted by chemoattractants from the injured microenvironment (for more information we suggest comprehensive reviews [18]).

To obtain more detailed information about the distribution, engraftment, and morphology of MSCs in target tissues, a study of semi-thin and ultrathin sections should be performed. Using TEM for the analysis of metal-NPs-labeled MSCs, we were able to determine the distribution and precise location of NPs-labeled MSCs in the tissues, as well as examine other cells that have taken up NPs and may be involved in the processes of metabolism and/or elimination from the body. With TEM, MSCs could be reliably identified from their typical ultrastructural features of undifferentiated cells containing metal electron-dense NPs [4,19]. Using light and TEM, MSCs were detected in the lungs and spleen of both healthy and cisplatin-treated mice in the greatest amount, whereas MSCs could also be detected by TEM in kidneys and intestines only in cisplatin-treated mice.

While MSCs found in the lungs retained their typical and original morphology, those found in the kidney, spleen, and intestine showed some signs of slight dysmorphism, including some changes in the nucleus and a slightly “compressed” appearance. Because the homing of MSCs is a complex, multistep process in which the settling cells may be activated, intravasate, migrate through the bloodstream, extravasate, migrate again, and undergo phenotypic changes [17], the dysmorphic appearance of MSCs in target tissues is to be expected. Conversely, such morphological changes were not observed in MSCs in the lungs, where their size probably allowed them to remain passively stuck (the effect of first passage through the lungs) and retain their original structure.

Based on TEM analysis, we assume that, beyond the passive accumulation in the lungs and to some extent in the spleen (in which the pathologic changes predominantly affected the filtering part, i.e., the red pulp), MSCs were still present in the most damaged tissues of the cisplatin-treated mice, i.e., kidney and intestine, whereas they were absent in these tissues in the control group (without organ damage). This observation is consistent with the immunohistochemical analysis using double immunostaining with CD105 and CD44 (cellular surface markers characteristically expressed by MSCs), as well as with our previous results using DiI labeling, and suggests that MSCs preferentially migrate into the damaged organs. However, the total number of MSCs found in kidneys and intestines on day four after administration was already low, because MSCs are normally eliminated from the body within 5 to 10 days after administration.

Further TEM analysis revealed that some NPs are also found outside MSCs, in significant amounts in the cytoplasm of epithelial cells of the proximal renal tubules and in Kupffer cells of the liver. Recent data from the literature have shown that many stem cells labeled with intracellular markers (including superparamagnetic iron oxide NPs) can be taken up by tissue-resident macrophages, complicating the interpretation of intracellular labels [20]. We did not find MSCs in the livers of the control and cisplatin mouse groups and there was an abundance of NPs found in the Kupffer cells of both groups, suggesting that Kupffer cells are likely involved in the metabolism and excretion of NPs via the hepatobiliary system, as also suggested by some other authors [21]. Similarly, renal proximal tubule epithelial cells might also be involved in the excretion of NPs via urine [22]. In both groups of mice, we found endosomes fully loaded with NPs and distributed in an ascending gradient from the basal (tubulointerstitial side) to the apical (urinary side) part of the cells. Because no retained MSCs were found in the renal tissue of the control group either, it can be hypothesized that the NPs probably originated from the systemic circulation, where they may have been secreted from decayed MSCs, endocytosed by the tubular epithelial cells, and finally excreted into the urine.

It has been confirmed that the cobalt ferrite NPs used in our study are nontoxic during 24-h exposure [10] and this was also the exposure time in our experiments. Therefore, we were not surprised to find no obvious abnormalities in the ultrastructure of MSCs or changes in the morphology of the tissues studied in the control mice. At present, however, knowledge about the effects of nanomaterials on human (stem) cells is still sparse and contradictory [23,24]. There are some reports of a possible effect of NPs (including cobalt ferrite NPs) on the enzyme activity and cell structure of MSCs, and large differences in responses to NPs have been reported between different cell types, NP concentrations, and exposure times [25,26,27]. Therefore, further in vivo studies are needed to determine the effects of NPs on different cell types, including stem cells, and to explore the optimal labeling strategy by NPs to avoid their potential toxic effects.

Altogether, our study indicates that NPs are useful markers to detect MSCs in tissues and to analyze their distribution in the body of animals. Metal cobalt ferrite NPs are visible under light microscopy and TEM, which allows us to perform a variant of correlative microscopy to detect MSCs labeled with NPs. This method could be beneficial in routine histopathology laboratories because it allows detection of MSCs with TEM, even in paraffin-embedded tissue, which is the most common form of biopsy tissue.

It is worth noting that our study is preliminary and has some important limitations. Because of the small number of animals used in the study, only qualitative conclusions can be drawn regarding the trafficking and distribution of labeled MSCs. Likewise, the number of animals used is not sufficient to draw definitive conclusions, so further comparative studies with a sufficiently large sample of animals are needed.

## 4. Materials and Methods

### 4.1. In Vivo Experiments

All procedures involving animals were approved by the National Ethics Committee and the Food Safety, Veterinary and Phytosanitary Administration of the Republic of Slovenia (approval number 34401-54/2012/5). Animal care and treatment were performed in accordance with institutional guidelines and international laws and directives (Directive 2010/63/EU on the protection of animals used for scientific purposes).

Acute kidney injury was induced with cisplatin as previously described [12]. In brief, experiments were performed on 8–12-week-old male BALB/cOlaHsd mice (Harlan, Bresso, Italy). Acute kidney injury models were induced by intraperitoneal injection of a single dose of 17 mg/kg cisplatin (Pliva-Teva, Zagreb, Croatia) dissolved in 0.9% saline (1 mg/2 mL). The dose of cisplatin was selected on the basis of the literature and preliminary experiments [11].

The mice were divided into two groups and received an intravenous injection of MSCs from the umbilical cord as follows: Group 1, 2 × 10^5^ MSCs (*n* = 2); Group 2, cisplatin + 2 × 10^5^ MSCs (*n* = 2). To study intra-organ localization, MSCs (at sixth passage) were labeled with CoFe_2_O_4_ NPs as described below and injected intravenously 24 h after cisplatin administration. On day 4 after cisplatin application, the mice were sacrificed with CO_2_.

At autopsy, solid organs (lungs, liver, intestine, spleen, kidneys) were harvested, weighed, and collected for histological, immunohistochemical, and TEM analyses to assess structural impairment and stem cell distribution.

### 4.2. Isolation, Cultivation, and Characterization of MSCs

Isolation and characterization of MSCs were performed as previously described [12]. Briefly, MSCs were isolated from Wharton’s jelly according to a standard protocol [27] approved by the National Ethics Committee (document number 134/01/11). Isolated MSCs clones were cultured in Dulbecco’s medium (DMEM; Sigma-Aldrich, Taufkirchen, Germany) with 10% fetal bovine serum (FBS; PAA Laboratories, Pasching, Austria) supplemented with 100 U penicillin (PAA Laboratories, Austria), 1.000 U streptomycin (PAA Laboratories, Austria), 2 mM L-glutamine (PAA Laboratories, Austria), Na-pyruvate (Gibco, Invitrogen, Carlsbad, CA, USA), and nonessential amino acids (Sigma-Aldrich, Germany). The MSCs clones were characterized for expression of CD13+, CD29+, CD44+, CD73+, CD90+, CD105+, CD14-, CD34-, CD45-, and HLA-DR-surface markers, as well as osteogenic, chondrogenic, and adipogenic differentiation according to the recommendations [19]. A mixture of three MSCs clones with the highest proliferation potential and homogeneous spindle-like morphology was used in animal experiments.

### 4.3. Labeling of MSCs with Metal NPs

To identify MSCs in animal tissues under light microscope and TEM, we labeled MSCs in vitro with cobalt ferrite (CoFe_2_O_4_) NPs (provided by the group of Mojca Pavlin, Faculty of Electrical Engineering, University of Ljubljana). The NPs were synthesized by the coprecipitation method as described by Bregar and colleagues [10]. Shortly after NPs were obtained, they were coated in situ with a water solution of polyacrylic acid (PAA) sodium salt, resulting in functionalized and long-term highly stable particles in the water solution with minimal leeching of cobalt, and with an average diameter of 33 nm in the water solution. NPs are proved to be non-toxic in physiological conditions over a broad range of pH values and prevented from extracellular or intracellular aggregation.

The day before infusion into the animals, MSCs were exposed to cobalt ferrite NPs for 24 h. In previous studies conducted by the supplier, cobalt ferrite NPs at a concentration of 100 µg/mL were found to be non-toxic and very stable in the short term [25,26]. After 24 h, MSCs were rinsed with fresh cell culture medium. The viability of the MSCs, as determined by a Trypan blue exclusion assay, was always >96%. MSCs with internalized NPs were then injected into animals.

### 4.4. Preparation of Paraffin, Semi-Thin, and Ultrathin Sections for Correlative Microscopy

After sacrificing the animals, the lungs, spleen, intestine, liver, and kidney of the cisplatin-treated and control mice were excised and fixed in 4% paraformaldehyde (Merck, Darmstadt, Germany), dehydrated in ethanol, and embedded in paraffin. The paraffin- embedded tissues were then cut into 4 µm thick sections and stained with hematoxylin and eosin. The sections were examined with a light microscope (10- to 600-fold magnification, Nikon, Tokyo, Japan) to assess the tissue damage caused by cisplatin administration and to find the MSCs loaded with NPs in the tissues.

Cisplatin-induced tissue injury was assessed semi-quantitatively in histological sections for the presence of apoptotic cells, mononuclear inflammatory cells (lymphocytes, macrophages, plasma cells), and injury to parenchymal cells (tubular epithelial cells, hepatocytes, pneumocytes in alveoli, enterocytes of the intestine). Tissue areas in paraffin sections with scattered cells containing brown particles in the cytoplasm, later identified as MSCs with internalized metal NPs, were punched from the paraffin block and processed for TEM. Randomly selected areas (2 mm^3^) from paraffin sections of organs in which cells with brown particles in the cytoplasm were not visible under the light microscope were punched from the paraffin block and processed for TEM.

Semi-thin and ultrathin sections were also prepared from paraffin-embedded tissue sections of the lungs, spleen, intestine, liver, and kidney cortex. After deparaffinization with xylene (J. T. Baker, Gliwice, Poland) and rehydration with ethanol at decreasing concentrations, tissue samples were placed overnight in 0.1 M Millonig’s phosphate buffer and processed the next day for TEM. First, tissue pieces were post-fixed in 1% OsO4 (Merck, Darmstadt, Germany) for 30 min, then dehydrated in graded concentrations of ethanol and later in propylene oxide 1-2-propylene oxide (Merck, Darmstadt, Germany) for 10 min each. The tissue pieces were then incubated in a mixture (1:1) of 1–2-propylene oxide and Epon 812 resin (Serva Electrophoresis, Heidelberg, Germany) for 20 min, embedded in 100% Epon, and polymerized overnight at 60 °C. The next day, semi-thin sections were cut using a Leica EM UC6 ultramicrotome and stained with Azure II staining solution (BDH Chemicals Ltd., Poole, England) to find MSCs with brown particles in the cytoplasm. Ultrathin sections (60 nm thick) were counterstained with 5% uranyl acetate (SPI Supplies, West Chester, PA, USA) and 3% lead citrate (Merck, Darmstadt, Germany) and examined in a transmission electron microscope JEM-1200 EXII (JEOL, Tokyo, Japan) at 60 kV.

### 4.5. Immunofluorescence Labeling and Detection of Immunolabeled MSCs

Paraffin sections of mouse tissue samples were deparaffinized and permeabilized with ice-cold 100% ethanol for 15 min at room temperature. After rinsing in phosphate-buffered saline (PBS; Merck, Germany), nonspecific labeling was blocked with 5% bovine serum albumin (BSA; Sigma-Aldrich, Germany) in PBS for 1 h at 37 °C. Primary mouse monoclonal antibodies to CD44 (1:200; MRQ-13, Cell Marque, Rocklin, CA, USA) and rabbit monoclonal antibodies to CD105 (1:100; ER274, Epitomics Inc., Pleasanton, CA, USA) were applied and incubated overnight at 4 °C. After rinsing in PBS, the corresponding secondary antibodies (1:400, goat anti-mouse Alexa Fluor 488, Invitrogen; goat anti-rabbit Alexa Fluor 555, Invitrogen, Carlsbad, CA, USA) were applied for 1 h at 37 °C. Positive controls were performed for CD44 on normal urothelium and for CD105 on normal vascular endothelium, as indicated in the antibody manufacturers’ technical specifications. Sections were mounted in Vectashield mounting medium with DAPI (Vector Laboratories, Burlingame, CA, USA) and examined with the AxioImager Z1 fluorescence microscope (Zeiss, Oberkochen, Germany).

## 5. Conclusions

Studies investigating stem cell homing, their cell–cell interactions, and their behavior in tissues after transplantation rely on proper tracking of the stem cells after administration. Cobalt ferrite NPs are not only biocompatible, but also have unique chemical and physical properties that allow transplanted cells to be tracked and imaged in real time using various imaging techniques, as well as ex vivo using light microscopy and TEM. When MSCs are labeled with NPs, histology together with ultrastructural features of MSCs should be used in combination with MR imaging to allow a better understanding of the role of MSCs in tissue regeneration and to enable continuous improvement of therapeutic and diagnostic methods.

## Figures and Tables

**Figure 1 ijms-23-08738-f001:**
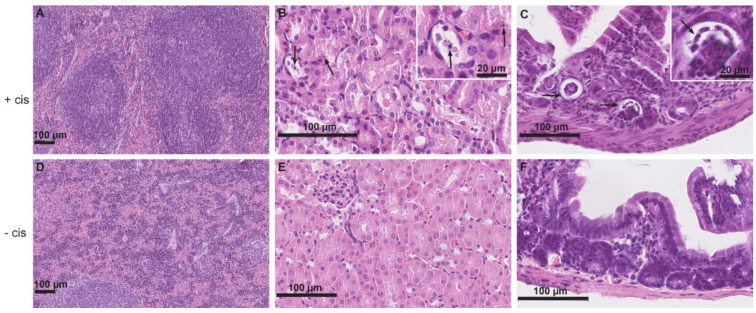
Representative micrographs of histopathologic changes of the spleen, kidney, and intestine in cisplatin-treated mice receiving MSCs labeled with NPs (+cis) (**A**–**C**) compared to control mice receiving MSCs labeled with NPs without cisplatin treatment (−cis) (**D**–**F**). (**A**) In the spleen, cisplatin-induced widening of the red pulp and blurring of the boundaries between white and red pulp is evident. (**B**) In the renal cortex, numerous apoptotic tubular epithelial cells (arrows) are found. In the upper right corner is an insert with apoptotic epithelial cells at higher magnification. (**C**) In the intestine, necrotic epithelial cells are present particularly in crypts (arrows). In the upper right corner is an insert with necrotic epithelial cells at higher magnification. (**D**–**F**) Normal parenchyma of the spleen, kidney, and intestine of control mice receiving MSCs labeled with NPs. Scale bars: 100 µm and 20 µm.

**Figure 2 ijms-23-08738-f002:**
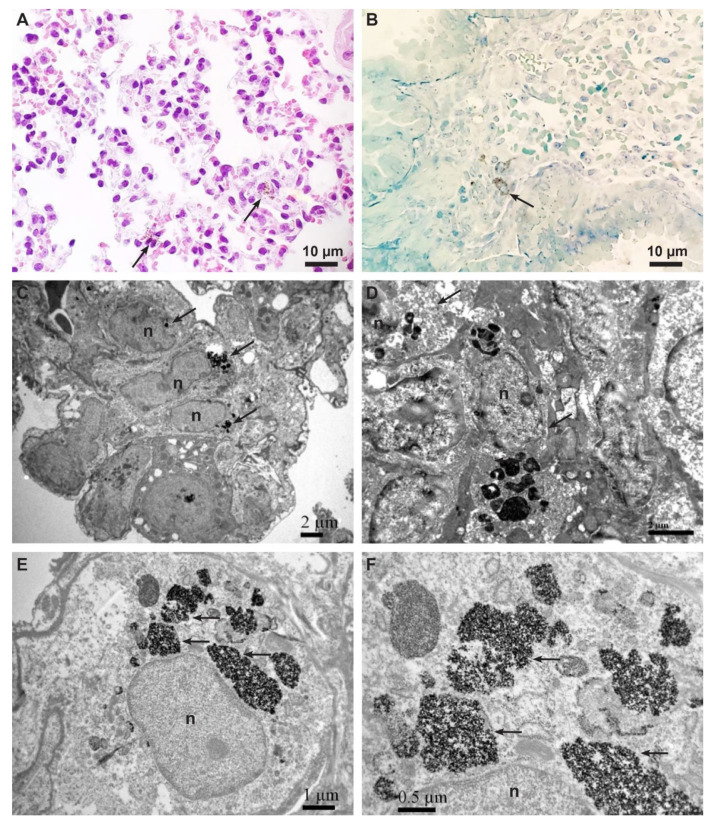
Representative micrographs of MSCs with internalized CoFe_2_O_4_. (**A**) Light micrograph of hematoxylin and eosin staining showing scattered cells containing brown particles inside the cytoplasm (arrows). (**B**) Micrograph of a semi-thin section stained with Azure II. Note the cell full of brown particles inside the cytoplasm (arrow). (**C**) TEM micrograph of three MSCs (arrows) containing electron-dense NPs in endosomes in the lungs of control mouse. Scale bar: 2 µm. (**D**) TEM micrograph of two MSCs (arrows) containing electron-dense NPs in endosomes in the spleen of cisplatin-treated mouse. Scale bar: 2 µm. (**E**) TEM micrograph of MSC containing electron-dense NPs in numerous endosomes (arrows) in the lungs of cisplatin−treated mouse. Scale bar: 1 µm. (**F**) Higher magnification of endosomes (from (**E**)) fully loaded with electron−dense NPs (arrows). Scale bar: 0.5 µm. Legend: n−nucleus.

**Figure 3 ijms-23-08738-f003:**
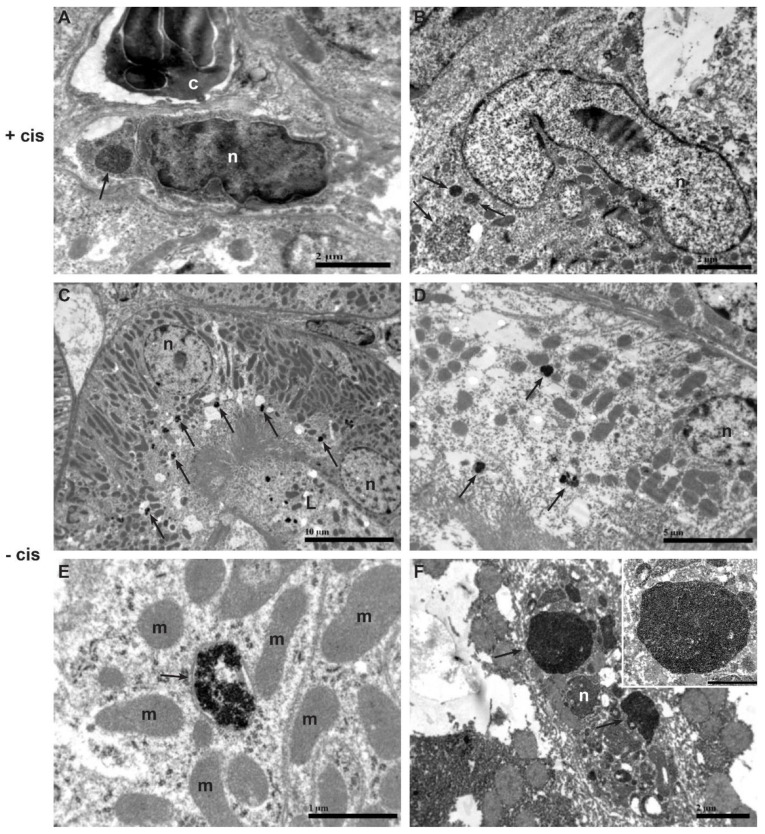
Representative TEM micrographs of intracellular localization of electron−dense NPs in the kidney and intestine of cisplatin-treated mouse (+cis) (**A**,**B**) and in the kidney and liver of control mouse (−cis) (**C**–**F**). (**A**) Electron-dense NPs (arrow) in endosome of MSC located in interstitium between peritubular capillary and tubular basement membrane. Scale bar: 2 µm. (**B**) Three endosomes containing electron−dense NPs (arrows) inside the cytoplasm of MSC in intestine. Scale bar: 2 µm. (**C**) Several small endosomes with electron-dense NPs (arrows) in apical cytoplasm of proximal tubular epithelial cells. Scale bar: 10 µm. (**D**) Endosomes with electron-dense NPs (arrows) in basal and apical cytoplasm of proximal tubular epithelial cell. Scale bar: 5 µm. (**E**) Higher magnification of endosome containing NPs (arrow) in proximal tubular epithelial cell. Scale bar: 1 µm. (**F**) Electron-dense NPs in endosomes (arrows) of Kupffer cell. Scale bar: 2 µm. Note a high magnification view of large endosome fully loaded with electron-dense NPs in the upper right corner of the figure. Scale bar: 1 µm. Legend: n−nucleus; c−cytoplasm of neutrophilic granulocyte; L−lumen of proximal tubule; m−mitochondria.

**Figure 4 ijms-23-08738-f004:**
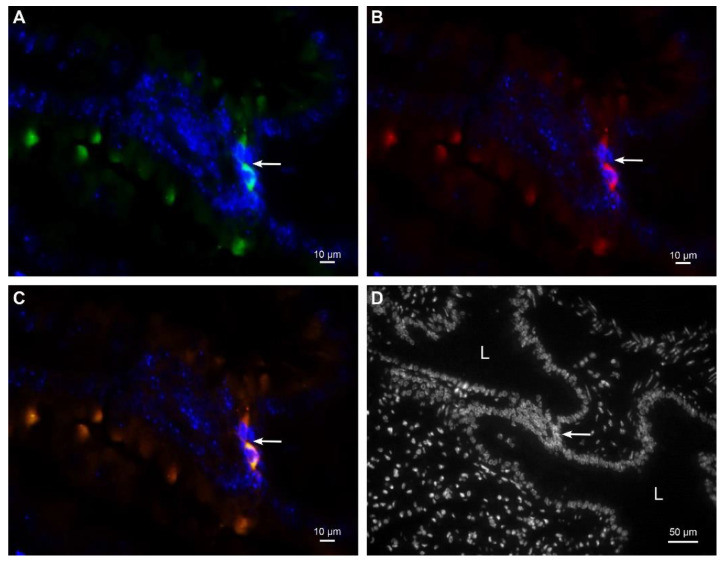
Representative images of immunofluorescent labeling against CD44 and CD105 in lungs. Two MSCs with positive immunoreaction against CD44 (green fluorescence) (**A**) and CD105 (red fluorescence) (**B**). The same two MSCs are orange−colored due to merged green fluorescence of CD44 and red fluorescence of CD105 confirming the colocalization of both surface markers (**C**). MSCs were located in the interstitium between alveoli ((**D**) nuclear staining only). Nuclei are stained with DAPI. Arrow denotes two MSCs. L−lumen of the alveolus.

**Table 1 ijms-23-08738-t001:** Localization and semi quantitative assessment of nanoparticles and MSCs containing nanoparticles by light and transmission electron microscopy.

Location of NPs and MSCs Containing NPs	Cisplatin-Treated Mice		Control Mice	
	LM	TEM	LM	TEM
Lungs (MSCs)	+++	+++	+++	+++
Spleen (MSCs)	+++	+++	+++	+++
Kidney (MSCs in interstitium)	−	+	−	−
Kidney (NPs in PTEC)	−	+	−	+
Intestine (MSCs)	−	+	−	−
Liver (NPs in Kupffer cells)	±	+	±	+
Liver (MSCs)	−	−	−	−

Legend: NPs—nanoparticles; LM—light microscopy; TEM—transmission electron microscopy; MSCs—mesenchymal stem cells; PTEC—proximal tubular epithelial cells. Semi quantitative assessment of nanoparticles in organs and cells: +++ nanoparticles visible, ++ few nanoparticles visible, + very few nanoparticles visible, ± suspicious nanoparticles visible, − no convincing nanoparticles visible.

## Data Availability

Not applicable.

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
