# Peer review of "Cobalt Ferrite Magnetic Nanoparticles for Tracing Mesenchymal Stem Cells in Tissue: A Preliminary Study"

_ijms, 2022, doi:10.3390/ijms23158738_

Round 1
Reviewer 1 Report
This article demonstrates the effectiveness of cobalt ferrite magnetic nanoparticles in labelling of mesenchymal stem cells and detecting their post-translational localisation in the mouse body.
The article is interesting, well-written and clear, but contains imperfections that should be corrected.
Figures 1 B and C - the cells indicated by the arrows are not clearly visible - please add inserts with a fragment of the microphotograph at higher magnification
line 173 - please change "Representative images of immunofluorescence labelling against..." to "Representative images of immunofluorescent labelling against..."
The authors report that they used only four mice for the study: two in each group. This is a very small number. Therefore, the presented article should be treated as a description of phenomenology and add in the title the phrase "Preliminary data".
Author Response
REVIEWER'S COMMENTS AND AUTHOR RESPONSES
Reviewer 1
The article is interesting, well-written and clear, but contains imperfections that should be corrected.
COMMENT 1. Figures 1 B and C - the cells indicated by the arrows are not clearly visible - please add inserts with a fragment of the microphotograph at higher magnification
Author response
We kindly thank the Reviewer for the valuable suggestion. Listening to the Reviewer's comment we have added inserts with indicated fragments of the micrograph at higher magnification and we hope that they will contribute to better visibility of the micrographs.
COMMENT 2. line 173 - please change "Representative images of immunofluorescence labelling against..." to "Representative images of immunofluorescent labelling against..."
Author response
Following the Reviewer's suggestion we have changed the text to Figure 4 as it was suggested.
COMMENT 3. The authors report that they used only four mice for the study: two in each group. This is a very small number. Therefore, the presented article should be treated as a description of phenomenology and add in the title the phrase "Preliminary data".
Author response
We appreciate the Reviewer's comment on the issue regarding the number of animals included in our study and we fully agree with the Reviewer that our study describes the phenomenon of the homing of MSCs after systemic infusion and the methods for tracing and identifying them. According to the Reviewer's suggestion we therefore adequately changed the title of our manuscript and is now as follows: Cobalt ferrite magnetic nanoparticles for tracing mesenchymal stem cells in tissue: a preliminary study.
Reviewer 2 Report
The work makes a good impression. However, the arsenal of methods and the methodological base do not imply serious interest for researchers. I would recommend improving the work, or try to submit an article to a less rated journal. 1) The title of the article should be changed and specified. Since only one type of nanoparticles is used. 2) There is no information about the method of obtaining nanoparticles and their physical properties. 3) Publication in the desired high-ranking journal requires statistical processing of the results. 4) At a minimum, PCR analysis of genes regulating cell death and survival after exposure to MSCs labeled with nanoparticles should be applied.
Author Response
REVIEWER'S COMMENTS AND AUTHOR RESPONSES
Reviewer 2
COMMENT 1. The title of the article should be changed and specified. Since only one type of nanoparticles is used.
Author response
We kindly thank the Reviewer for the comment, which we agree with so we adequately changed the title of our manuscript and is now as follows: Cobalt ferrite magnetic nanoparticles for tracing mesenchymal stem cells in tissue: a preliminary study.
COMMENT 2. There is no information about the method of obtaining nanoparticles and their physical properties.
Author response
We thank the Reviewer for this comment. We labeled MSCs in vitro with cobalt ferrite (CoFe2O4) NPs (provided by the group of Mojca Pavlin, Faculty of Electrical Engineering, University of Ljubljana) synthesized and characterized by the coprecipitation method and coated in situ with polyacrylic acid (PAA) so the size of particles is 33 nm (average diameter) in water solution. These NPs are chemically stable, the leeching of cobalt is minimal, are non-toxic in physiological conditions over a broad range of pH values, and prevented from extracellular or intracellular aggregation. All additional and detailed data can be found in our reference No. 10 (Bregar V. et al., Int J Nanomed 2013), where the authors precisely described the preparation protocol and the characteristics of NPs. We have modified the text about NPs so please find it as yellow highlighted text in paragraph 4.3. of the Material and Methods section in a revised version of our manuscript.
COMMENT 3. Publication in the desired high-ranking journal requires statistical processing of the results.
Author response
We appreciate the Reviewer's comment on this issue and we are aware of that, but our study is just a preliminary study (we have also changed the title of the manuscript to emphasize that), which describes a phenomenon of the homing of MSCs after systemic infusion in animals to study and understand the distribution and engraftment of exogenously injected MSCs. Our results are therefore strictly qualitative and semi-quantitative and do not intend to show quantitatively and statistically the differences inside the animal group or between the groups. With all due respect to the Reviewer’s comment, statistical analysis is thus unfeasible and unnecessary. Namely, we just wanted to detect injected MSCs inside the animal body/various organs so we had to find markers for MSCs and an accurate tracking method for an identification and a tracing of MSCs in tissues as described in our manuscript. We truly believe that the number of animals used is thus sufficient for the given purpose.
COMMENT 4. At a minimum, PCR analysis of genes regulating cell death and survival after exposure to MSCs labeled with nanoparticles should be applied.
Author response
With all due respect to the Reviewer’s opinion, suggested method is beyond the scope of our study. Namely, our goal was to find a way to identify and trace injected MSCs and analyse their distribution in animal body. To our opinion, these methods and acquired information about MSC engraftment are important and of great value for MSC-based therapeutic strategies in clinics in the future. The analysis of the impact of MSCs on tissue homeostasis and regeneration of damaged/pathological tissue is planned for future investigations and goes beyond the scope of this work.
Round 2
Reviewer 2 Report
The author's answers seem quite convincing to me. I like changing the title of the article to something less ambitious. However, the discussion should be somewhat rewritten in the light of the fact that the experiments carried out are largely pilot ones. Is it possible to improve the quality of Figure 4 (optional).
Author Response
Reviewer 2
The author's answers seem quite convincing to me. I like changing the title of the article to something less ambitious.
- However, the discussion should be somewhat rewritten in the light of the fact that the experiments carried out are largely pilot ones.
Author response
As suggested by the reviewer we have attempted to tone down the discussion by minimizing the strength of the conclusions and introducing a paragraph on limitations. Please find the text changes of the Discussion section marked up using the “Track Changes” function or highlighted in yellow in the new version of our manuscript.
- Is it possible to improve the quality of Figure 4 (optional).
Author response
According to the Reviewer’s suggestion we improved the resolution of Figure 4 and consequently also the quality of the image. Please find the new version of Figure 4.